# Irregular Work Hours and the Risk of Sleep Disturbance Among Korean Service Workers Required to Suppress Emotion

**DOI:** 10.3390/ijerph18041517

**Published:** 2021-02-05

**Authors:** Sehyun Yun, Minsuk Kim, Won-Tae Lee, Jin-Ha Yoon, Jong-Uk Won

**Affiliations:** 1Department of Occupational and Environmental Medicine, Severance Hospital, Yonsei University College of Medicine, Seoul 03722, Korea; yunsehyun@yuhs.ac (S.Y.); bluemsk89@yuhs.ac (M.K.); lewot20@yuhs.ac (W.-T.L.); 2The Institute for Occupational Health, Yonsei University College of Medicine, Seoul 03722, Korea; 3Graduate School of Public Health, Yonsei University College of Medicine, Seoul 03722, Korea; flyinyou@yuhs.ac; 4Department of Preventive Medicine, Yonsei University College of Medicine, Seoul 03722, Korea

**Keywords:** work schedule, irregular schedule, shift work, sleep disturbance

## Abstract

Although a necessity in a modern society, irregular work schedule can lead to sleep problems. We investigated the effect of work schedule irregularity on sleep disturbance of 17,846 Korean service workers using the fifth Korean Working Conditions Survey. The odds ratio (OR) and 95% confidence interval (CI) for sleep disturbance occurrence were calculated through a multiple logistic regression model. The adjusted ORs for moderate and severe sleep disturbances for those with irregular work hours were 2.11 (95% CI 1.90–2.33) and 3.10 (95% CI 2.62–3.66), respectively. Work schedule irregularity and emotion suppression at work showed synergistic effect on both moderate and severe sleep disturbances. Sleep disturbances can lead to brain function deterioration and work-related injuries; therefore, appropriate measures should be addressed for the vulnerable population.

## 1. Introduction

It is evident that traditional work hours are not sufficient to fully meet the needs of a 24 h modern society. Various working hours have been implemented to accommodate for 24 h services, production, or emergency medical care. These include shift schedule, rotating schedule, or irregular shift schedule involving irregular or unusual hours, completely different from traditional work hours [1]. Irregular work hours are prevalent in numerous professions, such as law enforcers, truck drivers, air traffic controllers, airline pilots, and healthcare professionals [2]. On average, about 15–30% of all workers are engaged in shift work in developed countries [3].

Although crucial to modern society, an irregular work schedule poses great threats to workers. Unfixed working hours can have significant negative impact on many aspects of workers’ health, including physical, psychological and psychosocial health, as well as workers’ job performance [4]. For example, nurses working night shifts show an increased risk of colorectal cancer [5], diabetes [6], immune function deterioration [7], and obesity [6]. A systematic review linked night shift work with loss of sleep, coronary heart disease, and breast and prostate cancers [8]. There are growing evidences that link not only nurses but all shift workers to numerous adverse health outcomes, including increased risk factors for cardiovascular disease [9,10,11], metabolic syndrome [12], and autoimmune hypothyroidism [13].

However, the most common side effect of shift work is related to sleep. It is estimated that about 32% of night workers, 10% of day workers, and 8–26% of rotating shift workers suffer from impaired sleep conditions [14]. Individuals with irregular work schedules often present misalignment of the circadian rhythm with their sleep–awake schedules. This discrepancy between the two cycles results in poor sleep quality, reduced sleep hours, and excessive day drowsiness [15,16]

Various work-related factors other than work schedule may influence sleep condition. Some studies showed that increases in work demands, including both physical and emotional workloads, lead to a higher risk of developing insomnia [17,18]. Lim et al. claimed that excessive emotional demands can lead to insomnia in Korean paid workers [19]. It has been shown that, for service workers, emotional demand is much higher when they are required to suppress emotion at workplace, because regulating emotion mandates mental effort, leading to emotional burnout [20,21].

Therefore, in this study, we attempted to analyze the combined effects of work schedule regularity and emotion suppression in workplace on sleep disturbance in Korean service workers. Our study can aid in detecting vulnerable populations and be a foundation in creating an employee-friendly work environment.

## 2. Materials and Methods

Data for the study were obtained from the fifth Korean Working Condition Survey (KWCS) (2017). The KWCS is a modified version of the European Working Condition Survey to best accommodate Korean working population. The fifth KWCS includes randomly selected 50,205 consenting working individuals over the age of 15 years. We used the following criteria to select a sample that best served the purpose of the study: (1) paid worker; (2) individual younger than 65 years old; (3) service sector worker; and (4) individual who completed all relevant survey questions. Following these criteria, unpaid workers (*n* = 20,097), people over the age of 65 years (*n* = 2817), non-service sector workers (*n* = 9436), and people who did not answer all relevant questions (*n* = 9) were removed, and a total of 17,846 paid service sector employees younger than 65 years old were included in the study.

### 2.1. Work Regularity

The KWCS asks whether the length of work hours is the same every week, whether the number of working days is the same every week, whether the weekly work shift is fixed, and whether the start and end time of each shift is fixed. “Regular work schedule” was defined as a variable with fixed daily working hours, fixed weekly working days, fixed weekly work shift, and fixed start and end time of each shift, as indicated by the answers on the fifth KWCS. Work schedules that featured irregularity in any of the criteria were considered irregular [22].

### 2.2. Emotion Suppression

The survey question “Do you have to suppress your emotion at work?” was used to determine the degree of emotion suppression for each worker. Participants were separated into three categories based on the degree of emotion suppression: rarely (“never” and “almost never”), sometimes (“sometimes”), and always (“always” and “almost always”).

### 2.3. Sleep Disturbance

The fifth KWCS includes three questions derived from the Insomnia Severity Index, a tool for insomnia assessment. The three items included in the fifth KWCS are “difficulty falling asleep”, “difficulty staying asleep”, and “extremely tired even after sleep”. The individuals were to check the frequency of each of the three sleep-related problems in the past 12 months. The answers are categorized as “every day”, “several times a week”, “several times a month”, “rarely”, and “never”. We assigned points to each answer (“every day” = 5 points, “several times a week” = 4 points, “several times a month” = 3 points, “rarely” = 2 points, “never” = 1 point) and summed them up to create an insomnia index score. A score above seven points was considered to be moderate sleep disturbance, and a score above 10 points was considered to be severe sleep disturbance.

### 2.4. Covariates

Information of the following covariates were also obtained from the fifth KWCS: sex, age, education level, and weekly working hours. Age was grouped into five categories (below 25 years, between 25 and 34 years, between 35 and 44 years, between 45 and 54 years and greater than or equal to 55 years). The level of education was classified as “did not graduate high school”, “did not graduate college”, and “graduated college”. Weekly working hours were divided into less than 40 h, 40 h to 52 h, and more than 52 h; the legal limit of weekly working hours is 52 h in Korea. The size of the workplace was categorized as small (1–49 employees), medium (50–249 employees), or large (more than or equal to 250 employees).

### 2.5. Statistical Analysis

A frequency analysis of sex, age, education level, weekly working hours, emotion suppression at work, and work schedule regularity was conducted to show the demographic and occupational characteristics of service workers. Chi-squared tests were used to compare socioeconomic characteristics and occupational status based on the presence of moderate and severe sleep disturbances. The odds ratio (OR) and 95% confidence intervals (95% CIs) on the presence of sleep disturbances based on work regularity were calculated using a fully adjusted multiple logistic regression model. Model A involved crude analysis, Model B controlled for individual characteristics such as gender, age, and education level, and Model C controlled for the factors in Model B along with the workplace environment factors, such as weekly working hours and the size of the workplace. We also stratified the data by weekly working hours to determine whether working hours affected the magnitude of work regularity on sleep disturbances. The degree of interaction between work regularity and emotion suppression on sleep disturbances was also analyzed through a logistic regression model. Interaction effect was estimated by calculating the odds ratio of each work type and emotion suppression degree combination to the baseline (regular work schedule + no emotion suppression at work). All *p*-values were two-tailed, and *p*-values less than 0.05 were considered statistically significant. KWCS utilizes weights variable on the basis of the economically active population to correctly represent the active population distribution. The survey weight was applied in all analysis. All analysis were conducted with SAS version 9.4 (SAS Institute, Cary, NC, USA).

## 3. Results

Table 1 shows the demographic and work environment characteristics of the study population. Out of the demographic characteristics, gender and education level showed statistically significant associations to both moderate and severe sleep disturbances, while age showed a statistically significant association to only moderate sleep disturbance. Female service workers were more likely to suffer from moderate and severe sleep disturbances than male service workers (9.3% males vs. 11.6% females for moderate sleep disturbance and 2.8% males vs. 3.8% females for severe sleep disturbance). The prevalence of moderate sleep disturbance was the highest for individuals who did not graduate high school (13.3%) followed by those who did not graduate college (11.8%), and those who graduated college (9.4%), with *p* < 0.001. The prevalence of severe sleep disturbance was the highest for individuals who did not graduate high school (7.0%), followed by those who did not graduate college (3.8%), and those who graduated college (2.8%), with *p* < 0.001. People older than 55 years showed the highest prevalence of moderate sleep disturbance (12.8%), followed by people with age 45–54 years (11.4%), people with age 25–34 years (10.4%), people with age less than 25 years (9.7%), and people with age 35–44 years (9.6%), with *p* = 0.0019.

Among the work environment characteristics, weekly working hours, emotion suppression at work, and work schedule were found to be associated with sleep disturbances. The proportions of moderate and severe sleep disturbances were the highest for weekly working hours less than 40 h (12.1% and 5.7%, respectively), followed by greater than 52 h (11.5% and 3.2%), and between 40 and 52 h (10.4% and 3.1%), with *p* = 0.0295 and *p* < 0.001, respectively. Those who always have to suppress emotion at work showed the highest prevalence for moderate and severe sleep disturbances (12.6% and 4.4%, respectively), followed by those who sometimes suppress emotion (10.2% and 2.9%), and those who rarely suppress emotion (6.9% and 1.9%), with *p* < 0.001 for both. Employees with irregular work schedule showed higher prevalence for both moderate and severe sleep disturbances (17.5% and 7.3%, respectively) than those who have regular work schedules (9.0% and 2.4%), with *p* < 0.001 for both.

Table 2 and Table 3 display the effect of work schedule regularity on moderate and severe sleep disturbances. Model A depicts the crude model, Model B is controlled for individual characteristics such as gender, age, and education level, and Model C is controlled for individual and work environment factors such as weekly working hours and the size of the workplace. The overall irregular work schedule had an adjusted OR of 2.11 (95% CI 1.90–2.33) compared to the regular work schedule for the risk of moderate sleep disturbance. Adjusted ORs were 2.35 (95% CI 2.10–2.62), 2.82 (95% CI 2.48–3.22), 2.01 (95% CI 1.80–2.29), and 3.18 (95% CI 2.71–3.71) for unfixed daily working hours, unfixed weekly working days, unfixed weekly work shift, and unfixed start and end time of each shift, respectively, when compared to their fixed counterparts. According to Table 3, the overall irregular work schedule had an adjusted OR of 3.10 (95% CI 2.62–3.66) compared to the regular work schedule for the risk of severe sleep disturbance. Adjusted ORs were 3.86 (95% CI 3.25–4.57), 4.96 (95% CI 4.11–5.98), 3.21 (95% CI 2.68–3.83), and 5.28 (95% CI 4.25–6.52) for unfixed daily working hours, unfixed weekly working days, unfixed weekly work shift, and unfixed start and end time of each shift, respectively, when compared to their fixed counterparts.

Table 4 and Table 5 depict the results of the stratification analysis based on weekly working hours. The risk of moderate sleep disturbance associated with irregular work schedule compared to that of regular work schedule were higher in a sub-group with less than, or equal to, 52 weekly working hours (OR 2.20, 95% CI 1.97–2.46 for “working hours ≤52 h/week” and OR 1.76, 95% CI 1.28–2.38 for “working hours >52 h/week”). This was also true for the risk of severe sleep disturbance (OR 3.22, 95% CI 2.71–3.83 for “working hours ≤52 h/week” and OR 2.64, 95% CI 1.56–4.40 for “working hours >52 h/week”).

Figure 1 and Figure 2 delineate the interaction effect between work schedule regularity and emotion suppression at work on moderate and severe sleep disturbances. According to Figure 1, the risk of moderate sleep disturbance increases as the magnitude of emotion suppression increases in both regular and irregular work schedules. When compared to the reference group (“regular work schedule and no emotion suppression”), the OR of moderate sleep disturbance for “regular work schedule and always emotion suppression” was 1.73 (95% CI 1.44–2.09), the OR for “irregular work schedule and no emotion suppression” was 1.85 (95% CI 1.34–2.52), and the OR for “irregular work schedule and always emotion suppression” was 4.19 (95% CI 3.43–5.13), which suggests the presence of a synergistic effect between work schedule regularity and emotion suppression on sleep disturbance. All other work-related factors showed synergistic effects with emotion suppression on moderate sleep disturbance.

According to Figure 2, the risk of severe sleep disturbance also increases as the magnitude of emotion suppression increases in both regular and irregular work schedules. When compared to the reference group (“regular work schedule and no emotion suppression”), the OR of severe sleep disturbance for “regular work schedule and always emotion suppression” was 1.41 (95% CI 1.00–2.03), the OR for “irregular work schedule and no emotion suppression” was 1.53 (95% CI 0.82–2.72), and the OR for “irregular work schedule and always emotion suppression” was 6.39 (95% CI 4.59–9.10), which indicates the presence of a synergistic effect between work schedule regularity and emotion suppression on sleep disturbance. All other work-related factors showed synergistic effects with emotion suppression on severe sleep disturbance.

## 4. Discussion

According to the study, an irregular work schedule was found to be associated with both moderate and severe sleep disturbances. Another factor associated with sleep disturbances was the magnitude of emotion suppression at work. The more frequent the employees were required to suppress emotion, the more likely they were to experience disturbances in sleep. It should also be noted that these two factors interact with each other and increase the risk of sleep disturbance. Our analysis indicates that an irregular work schedule and the degree of emotion suppression show a synergistic interaction. In other words, when employees with irregular work schedules are required to suppress emotion frequently, they present a much higher risk of having both moderate and severe sleep disturbances.

In Korea, the prevalence of insomnia symptoms is reported to be 17–23%, and the prevalence of insomnia diagnosis according to DSM-IV (Diagnostic and Statistical Manual - IV) is about 5% [23,24]. The numbers are slightly higher than the prevalence of sleep disturbances in our study, assuming that moderate sleep disturbance reflects insomnia symptoms and severe sleep disturbance indicates insomnia diagnosis (9.3% males vs. 11.6% females for moderate sleep disturbance and 2.8% males vs. 3.8% females for severe sleep disturbance). Old age is a major risk factor for sleep problems [25,26]; therefore, the lower prevalence may be due to the fact that individuals older than 65 were excluded from our study to best replicate the working population.

As expected, the risks of sleep disturbances were higher for employees with irregular work schedules (adjusted OR 2.11; 95% CI 1.90–2.33 for moderate sleep disturbance and adjusted OR 3.10; 95% CI 2.62–3.66 for severe sleep disturbance). Sleeping is mainly regulated by both homeostatic processes and the circadian rhythm [27]. In humans, the homeostatic process is in charge of sleep maintenance through the regulation of cortisol levels, while the circadian process controls the timing of sleep through melatonin levels [28]. Individuals with traditional working schedules show minimum cortisol levels before their regular bedtime and maximum levels approximately 30 min after waking up [29,30]. On the other hand, melatonin is at a maximum before regular bedtime and is at a minimum during the day when there is an environmental light signal present [31,32]. This regulation of cortisol and melatonin is crucial in sleep regulation.

Shift workers, however, frequently experience circadian rhythm desynchrony which is similar to those with jet-lag. Shift workers often experience light signals at night; therefore, they show circadian rhythm disruption and display a lack of correlation between cortisol and melatonin levels [33]. Cortisol levels during sleep for shift workers are higher than those of daytime workers, and a reduction in cortisol levels when awake for shift workers is smaller than that of daytime workers. Melatonin in shift workers is not at the highest during sleep, which is unlike that of day time workers. This imbalance in hormones creates sleep problems. Studies show that 32% of shift workers show symptoms of insomnia or excessive sleepiness, which is much higher than 18% of daytime workers [34].

Interestingly, magnitude of the risks of moderate and severe sleep disturbances is decreased when weekly working hours exceed 52 h (OR 2.20; 95% CI 1.97–2.46 for ≤52 h and OR 1.76; 95% CI 1.28–2.38 for >52 h in moderate sleep disturbance and OR 3.22; 95% CI 2.71–3.83 for ≤52 h and OR 2.64; 95% CI 1.56–4.40 for >52 h in severe sleep disturbance). This is because long working hours are associated with insomnia [35,36]. Working long hours reduces the amount of time to sleep [37]. Furthermore, people need more time to recuperate from long working hours [38,39]. Employees with long working hours cannot afford to engage in healthy lifestyle habits [35]. In other words, people who work more than 52 h per week are at risk of insomnia, regardless of their work schedules. As a result, the effect of work schedule irregularity on sleep disturbance compared with that of a regular work schedule is subdued.

As shown in Figure 1 and Figure 2, there is an ascending trend between the degree of emotion suppression and the risk of moderate and severe sleep disturbances in all work schedule types. Suppressing emotion increases the risk of mental fatigue [40]. Regulating emotion creates mental exhaustion, which results in heavy emotional burdens and a disruption of psychological health [21,41]. Increased emotional burden can activate stress systems, including the hypothalamic–pituitary–adrenal (HPA) axis and sympathetic nervous system [19]. Overactivation of the HPA axis results in increased level of cortisol. When cortisol levels are high, it binds to glucocorticoid receptors and stimulates periventricular nuclei, which ultimately results in increased norepinephrine levels [42]. Norepinephrine increases sleep EEG (electroencephalogram) frequency, which is associated with wakefulness. Increased HPA activity also induces sleep fragmentation, which severely lowers sleep quality [43]. Therefore, increased emotional burden due to emotion suppression at work increases the risk of sleep disturbance.

It should be noted that a synergistic interaction exists between work schedule irregularity and emotion suppression at work on sleep disturbances. As shown in Figure 1, the OR for moderate sleep disturbance for employees with irregular work schedules who always suppress emotion are much higher than the sum of ORs for moderate sleep disturbance for employees with either an irregular work schedule or always suppress emotion (OR 4.19 for the combined and OR 3.58 for the sum). The same is true for severe sleep disturbance (OR 6.39 for the combined and OR 2.94 for the sum). Both work schedule irregularity and emotion suppression disrupt sleep through the deregulation of sleep-related hormones, especially cortisol. Irregular work hours result in higher cortisol levels when asleep and reduced cortisol levels when awake. Increased emotional burden due to emotion suppression also results in increased cortisol levels, which leads to an increase in norepinephrine levels, and ultimately an increase in wakefulness. Our study shows that interplay between two factors that creates desynchrony in sleep-related hormones can severely increase the risk of developing sleep problems.

There are a number of limitations to be addressed. First, this study was a cross-sectional study based on survey questions, and thus no causal relationship between work schedule irregularity and sleep disturbance can be investigated. A further longitudinal study is needed to discover any causal relationships. Moreover, we arbitrarily defined moderate and severe sleep disturbances based on survey questions. However, the prevalence of moderate and severe sleep disturbances were similar to that of insomnia symptoms and insomnia diagnosis according to DSM-IV. Possible confounders for sleep disturbance, such as medical conditions, presence of children at home, and workload, were not considered. However, this study was based on a well-established population-based study with high statistical power. The results of the study address the possible risk factors for sleep disturbance.

## 5. Conclusions

We showed that both work schedule regularity and emotion suppression at work influence sleep quality; irregular working hours and constant emotion suppression increase the risk of sleep disturbance. More importantly, a synergistic interaction exists between irregular work schedules and always suppressing emotion at work on inducing sleep problems. In other words, employees with irregular working hours and are required to suppress emotion are much more likely to experience sleep disturbance. Therefore, appropriate measures should be addressed for the vulnerable population.

## Figures and Tables

**Figure 1 ijerph-18-01517-f001:**
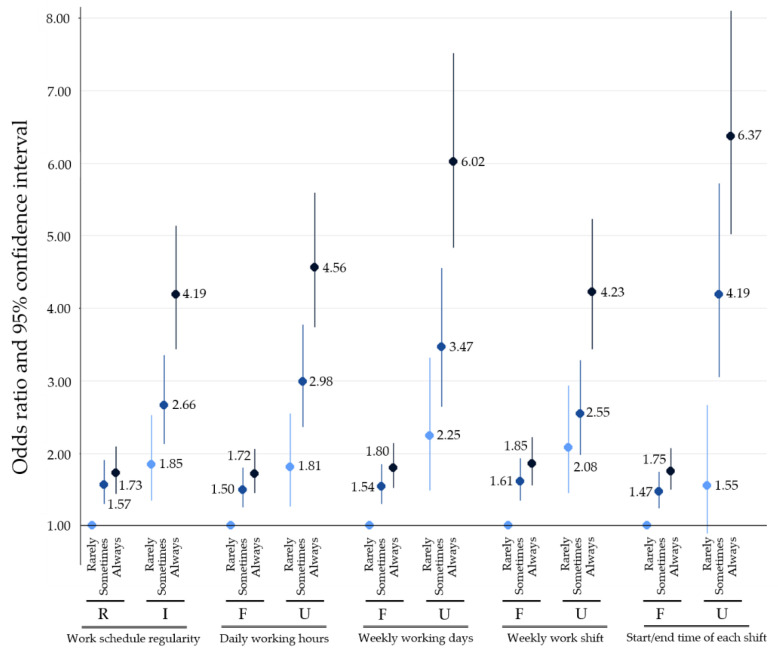
Odds ratios and 95% confidence intervals for moderate sleep disturbance according to work schedule regularity and frequency of emotion suppression at work. “R” stands for regular and “I” stands for irregular. “F” stands for fixed and “U” stands for unfixed. Regular work schedule indicates a schedule with fixed daily working hours, weekly working days, weekly work shift, and start and end time of each shift.

**Figure 2 ijerph-18-01517-f002:**
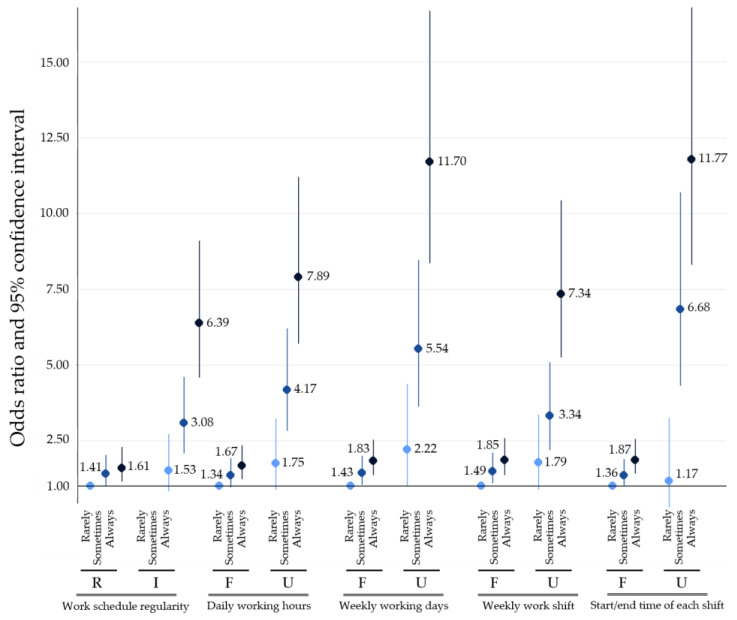
Odds ratios and 95% confidence intervals for severe sleep disturbance according to work schedule regularity and frequency of emotion suppression at work. “R” stands for regular and “I” stands for irregular. “F” stands for fixed and “U” stands for unfixed. Regular work schedule indicates a schedule with fixed daily working hours, weekly working days, weekly work shift, and start and end time of each shift.

**Table 1 ijerph-18-01517-t001:** Basic characteristics of the study population based on sleep disturbances. Chi-squared tests were used to compare socioeconomic characteristics and occupational status based on the presence of moderate and severe sleep disturbances.

Parameters	Total	Moderate Sleep Disturbance, No. (%)			Severe Sleep Disturbance, No. (%)		
		Yes	No	*p*-Value	Yes	No	*p*-Value
Gender				<0.001			<0.001
Male	6804	636 (9.3)	6168 (90.7)		193 (2.8)	6611 (97.2)	
Female	11,042	1281 (11.6)	9761 (88.4)		419 (3.8)	10,623 (96.2)	
Age (years)				0.0019			0.0966
<25	958	93 (9.7)	865 (90.3)		31 (3.2)	927 (96.8)	
25–34	4029	419 (10.4)	3610 (89.6)		137 (3.4)	3892 (96.6)	
35–44	5300	509 (9.6)	4791 (90.4)		156 (2.9)	5144 (97.1)	
45–54	5041	573 (11.4)	4468 (88.6)		186 (3.7)	4855 (96.3)	
≥55	2518	323 (12.8)	2195 (87.2)		102 (4.1)	2416 (95.9)	
Education Level				<0.001			<0.001
<High school	498	66 (13.1)	432 (86.7)		35 (7.0)	463 (93.0)	
<College	9196	1081 (11.8)	8115 (88.2)		348 (3.8)	8848 (96.2)	
≥College	8152	770 (9.4)	7382 (90.6)		229 (2.8)	7923 (97.2)	
Weekly working hours				0.0295			<0.001
<40	2128	258 (12.1)	1870 (87.9)		121 (5.7)	2007 (94.3)	
40–52	13,731	1430 (10.4)	12,301 (89.6)		428 (3.1)	13,303 (96.9)	
>52	1987	229 (11.5)	1758 (88.5)		63 (3.2)	1924 (96.8)	
Work size				0.1655			0.1394
Small	12,096	1336 (11.0)	10,760 (89.0)		456 (3.8)	11,640 (96.2)	
Medium	2418	244 (10.1)	2174 (89.9)		65 (2.7)	2353 (97.3)	
Large	3332	337 (10.1)	2995 (89.9)		91 (2.7)	3241 (97.3)	
Emotion suppression at work				<0.001			<0.001
Rarely	3023	209 (6.9)	2814 (93.1)		57 (1.9)	2966 (98.1)	
Sometimes	6739	686 (10.2)	6053 (89.8)		197 (2.9)	6542 (97.1)	
Always	8084	1022 (12.6)	7062 (87.4)		358 (4.4)	7726 (95.6)	
Work schedule *				<0.001			<0.001
Regular	14,225	1282 (9.0)	12,943 (91.0)		347 (2.4)	13,878 (97.6)	
Irregular	3621	635 (17.5)	2986 (82.5)		265 (7.3)	3356 (92.7)	
Daily working hours				<0.001			<0.001
Fixed	15,189	1403 (9.2)	13,786 (90.8)		374 (2.5)	14,815 (97.5)	
Not fixed	2657	514 (19.3)	2143 (80.7)		238 (9.0)	2419 (91.0)	
Weekly working days				<0.001			<0.001
Fixed	16,397	1576 (9.6)	14,821 (90.4)		432 (2.6)	15,965 (97.4)	
Not fixed	1449	341 (23.5)	1108 (76.5)		180 (12.4)	1269 (87.6)	
Weekly work shift				<0.001			<0.001
Fixed	15,514	1493 (9.6)	14,021 (90.4)		418 (2.7)	15,096 (97.3)	
Not fixed	2332	424 (18.2)	1908 (81.8)		194 (8.3)	2138 (91.7)	
Start and end time of each shift				<0.001			<0.001
Fixed	16,948	1677 (9.9)	15,271 (90.1)		482 (2.8)	16,466 (97.2)	
Not fixed	898	240 (26.7)	658 (73.3)		130 (14.5)	768 (85.5)	
Total	17,846	1917 (10.7)	15,929 (89.3)		612 (3.4)	17,134	

* Work schedule is regular when the daily working hours, weekly working days, weekly work shift, and start and end time of each shift are all fixed.

**Table 2 ijerph-18-01517-t002:** Odds ratio and 95% confidence intervals for moderate sleep disturbance.

Variables	Model A		Model B		Model C	
	ORs	95% CIs	ORs	95% CIs	ORs	95% CIs
Work schedule *						
Regular	1.00		1.00		1.00	
Irregular	2.15	1.94–2.38	2.11	1.90–2.34	2.11	1.90–2.33
Daily working hours						
Fixed	1.00		1.00		1.00	
Not fixed	2.36	2.11–2.63	2.35	2.10–2.62	2.35	2.10–2.62
Weekly working days						
Fixed	1.00		1.00		1.00	
Not fixed	2.89	2.53–3.30	2.83	2.49–3.23	2.82	2.48–3.22
Weekly work shift						
Fixed	1.00		1.00		1.00	
Not fixed	2.09	1.85–2.35	2.04	1.81–2.30	2.03	1.80–2.29
Start and end time of each shift						
Fixed	1.00		1.00		1.00	
Not fixed	3.32	2.84–3.88	3.18	2.71–3.72	3.18	2.71–3.71

* Work schedule is regular when the daily working hours, weekly working days, weekly work shift, and start and end time of each shift are all fixed. Model A: crude model; Model B: adjusted for gender, age, and education level; Model C: Model B + adjusted for emotion suppression at work, weekly working hours, and workplace size. OR: odds ratio; CI: confidence interval.

**Table 3 ijerph-18-01517-t003:** Odds ratio and 95% confidence intervals for severe sleep disturbance.

Variables	Model A		Model B		Model C	
	ORs	95% CIs	ORs	95% CIs	ORs	95% CIs
Work schedule *						
Regular	1.00		1.00		1.00	
Irregular	3.15	2.68–3.72	3.10	2.63–3.65	3.10	2.62–3.66
Daily working hours						
Fixed	1.00		1.00		1.00	
Not fixed	3.90	3.29–4.61	3.90	3.29–4.61	3.86	3.25–4.57
Weekly working days						
Fixed	1.00		1.00		1.00	
Not fixed	5.24	4.36–6.28	5.13	4.26–6.16	4.96	4.11–5.98
Weekly work shift						
Fixed	1.00		1.00		1.00	
Not fixed	3.28	2.74–3.90	3.21	2.69–3.83	3.21	2.68–3.83
Start and end time of each shift						
Fixed	1.00		1.00		1.00	
Not fixed	5.78	4.69–7.09	5.52	4.46–6.78	5.28	4.25–6.52

* Work schedule is regular when the daily working hours, weekly working days, weekly work shift, and start and end time of each shift are all fixed. Model A: crude model; Model B: adjusted for gender, age, and education level; Model C: Model B + adjusted for emotion suppression at work, weekly working hours, and workplace size. OR: odds ratio; CI: confidence interval.

**Table 4 ijerph-18-01517-t004:** Odds ratios and 95% confidence intervals for moderate sleep disturbance based on working hour stratification analysis.

Variables	Working Hours ≤52 h/Week		Working Hours >52 h/Week	
	ORs	95% CIs	ORs	95% CIs
Work schedule *				
Regular	1.00		1.00	
Irregular	2.20	1.97–2.46	1.76	1.28–2.38
Daily working hours				
Fixed	1.00		1.00	
Not fixed	2.40	2.13–2.70	2.04	1.45–2.83
Weekly working days				
Fixed	1.00		1.00	
Not fixed	2.99	2.60–3.43	2.22	1.43–3.34
Weekly work shift				
Fixed	1.00		1.00	
Not fixed	2.21	1.96–2.51	1.25	0.84–1.83
Start and end time of each shift				
Fixed	1.00		1.00	
Not fixed	3.37	2.85–3.97	2.95	1.78–4.75

* Work schedule is regular when the daily working hours, weekly working days, weekly work shift, and start and end time of each shift are all fixed. All were adjusted for individual characteristics (gender, age, and education level) and work environment factors (emotion suppression at work and workplace size).

**Table 5 ijerph-18-01517-t005:** Odds ratios and 95% confidence intervals for severe sleep disturbance based on working hour stratification analysis.

Variables	Working Hours ≤52 h/Week		Working Hours >52 h/Week	
	ORs	95% CIs	ORs	95% CIs
Work schedule *				
Regular	1.00		1.00	
Irregular	3.22	2.71–3.83	2.64	1.56–4.40
Daily working hours				
Fixed	1.00		1.00	
Not fixed	3.96	3.31–4.72	3.39	1.96–5.72
Weekly working days				
Fixed	1.00		1.00	
Not fixed	5.51	4.54–6.66	3.12	1.55–5.80
Weekly work shift				
Fixed	1.00		1.00	
Not fixed	3.51	2.92–4.22	1.62	0.81–2.98
Start and end time of each shift				
Fixed	1.00		1.00	
Not fixed	5.95	4.78–7.37	4.29	2.00–8.41

* Work schedule is regular when the daily working hours, weekly working days, weekly work shift, and start and end time of each shift are all fixed. All were adjusted for individual characteristics (gender, age, and education level) and work environment factors (emotion suppression at work and workplace size).

## Data Availability

Publicly available datasets were analyzed in this study. This data can be found here: [http://www.kosha.or.kr/ (accessed on 7 January 2021)].

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
