# Peer review of "Irregular Work Hours and the Risk of Sleep Disturbance Among Korean Service Workers Required to Suppress Emotion"

_ijerph, 2021, doi:10.3390/ijerph18041517_

Round 1
Reviewer 1 Report
Introduction
- Here, the authors suggest a causal link between sleep problems and work-related injuries via brain function: “Sleep problems in workers can easily cause work-related injuries. Many studies have indicated that sleep problems induce decline in brain function. This deterioration can result in working memory failure, executive function decline and shortened sustained attention [22]. Damaged brain function leads to increased risk of work injuries and workplace accidents. [23]. I don’t understand why injuries and brain function is brought into the introduction, and again in the conclusion, as the paper is not about these topics. Also, if it should be mentioned at all, it is referred to ‘many studies’, but only one is mentioned, ref 22, and this cover memory function in adolescents. It is not clear how, based on these references, that damaged brain function leads to increased risk of injuries. Please revise.
- The objective is defined as “analyzing the effect of work schedule regularity and emotion suppression at workplace on sleep disturbance in Korean service workers.” As three variables are included, I suggest that one or two hypotheses are formulated.
Methods
- From a European perspective, 15 year old workers are very young. It may be different in Korea. However, a sensitivity analysis excluding workers younger than e.g. 18 years may increase the external validity of the results.
- Work regularity needs some more details in explanation. Upon later reading table 2 and 3, I realize that work regularity is not one variable made up of the listed parameters. Instead, each of five parameters are entered separately into the statistical model. This is not comprehensible now and must be better explained. It would strengthen the paper if a reference was provided explaining the rationale for these particular questions. It is also not clear whether night work is included in the fixed schedule.
- Was there any aspect of self-rostering in the work scheduling, as this is supposedly benefitial for work-life balance and potentially sleep.
- Please provide a reference for the emotion suppression questions, or if these were self-made, state so.
- Please provide a reference for the insomnia severity index and a rationale for considering only three questions, the cut-off selected, etc. Using standardized questionnaires, measurement scales and cut-offs make comparison with other studies more relevant.
- Statistics: please explain in more detail the statistical tests used for calculating odds ratios in table 2-3 and figure 1-2. Particularly, the interaction analysis including emotional suppression needs more explanation.
Results
- Table 1: Please provide statistical test used in a footnote. Since table includes also statistical results, heading should reflect this.
- Regarding the statement: “Out of the demographic characteristics, gender and education level showed statistically significant correlations to moderate and severe sleep disturbances”. In my opinion, ‘association’ rather than ‘correlation’ is more correct.
- Table 2 and 3: It is not clear whether the Work schedule variable is fixed only if the other four variables are all fixed or not. See also comments on statistics section.
- Figure 1 and 2: I honour the authors for their effort in explaining these complex associations in a meaningful way.
Discussion
- Please start the discussion with a brief summary of the key results.
- I do not follow the string of arguments regarding why long working hours is associated with less sleep disturbance than shorter working hours. Please explain these thoughts in some more detail.
- Suggested biological mechanisms is fine, but somewhat less detail may be given to these aspects, as none of these variables were measured.
Conclusion: In my opinion, the second last sentence should be omitted. As mentioned before, the mention of brain fu
Reviewer 2 Report
This is a large-scale survey about sleep disturbance and the work schedule regularity and the emotional suppression among oriental population. This is a well-design and well-written manuscript. But I have still some minor concerns about this manuscript as following.
1.In table 1 the authors divided study population into part 1-moderate sleep disturbance and non-moderate sleep disturbance group, and the part 2 is divided into severe sleep disturbance and non-severe sleep disturbance. The data is vague in this grouping, and why not divide into normal, mild, moderate and severe group about sleep disturbance.? This should be clearer for data presentation. And if that's so, the table 2 and table 3 need to be re-analyzed again for univariate and multiple variate logistical regression.
2.Line 100 -Is this summation of each different question adequate? Because these figures are NOT continuous variables.
3.Line 157-Lack of percentage for all figures in this table 1.
Round 2
Reviewer 2 Report
The authors have replied all my questions and revised in the revision manuscript.
Author Response
Thank you for your thorough comments. Your comments helped us to write a better paper.